# Streamlining Variational Inference for Constraint Satisfaction Problems

**Aditya Grover, Tudor Achim, Stefano Ermon**
Computer Science Department
Stanford University
{adityag, tachim, ermon}@cs.stanford.edu

## Abstract

Several algorithms for solving constraint satisfaction problems are based on survey propagation, a variational inference scheme used to obtain approximate marginal probability estimates for variable assignments. These marginals correspond to how frequently each variable is set to true among satisfying assignments, and are used to inform branching decisions during search; however, marginal estimates obtained via survey propagation are approximate and can be self-contradictory. We introduce a more general branching strategy based on streamlining constraints, which sidestep hard assignments to variables. We show that streamlined solvers consistently outperform decimation-based solvers on random $k$-SAT instances for several problem sizes, shrinking the gap between empirical performance and theoretical limits of satisfiability by $16.3\%$ on average for $k = 3, 4, 5, 6$.

## 1  Introduction

Constraint satisfaction problems (CSP), such as boolean satisfiability (SAT), are useful modeling abstractions for many artificial intelligence and machine learning problems, including planning [13], scheduling [27], and logic-based probabilistic modeling frameworks such as Markov Logic Networks [30]. More broadly, the ability to combine constraints capturing domain knowledge with statistical reasoning has been successful across diverse areas such as ontology matching, information extraction, entity resolution, and computer vision [15, 4, 32, 29, 33]. Solving a CSP involves finding an assignment to the variables that renders all of the problem's constraints satisfied, if one exists. Solvers that explore the search space exhaustively do not scale since the state space is exponential in the number of variables; thus, the selection of branching criteria for variable assignments is the central design decision for improving the performance of these solvers [5].

Any CSP can be represented as a factor graph, with variables as nodes and the constraints between these variables (known as clauses in the SAT case) as factors. With such a representation, we can design branching strategies by inferring the *marginal probabilities* of each variable assignment. Intuitively, the variables with more extreme marginal probability for a particular value are more likely to assume that value across the satisfying assignments to the CSP. In fact, if we had access to an oracle that could perform exact inference, one could trivially branch on variable assignments with non-zero marginal probability and efficiently find solutions (if one exists) to hard CSPs such as SAT in time linear in the number of variables. In practice however, exact inference is intractable for even moderately sized CSPs and approximate inference techniques are essential for obtaining estimates of marginal probabilities.

Variational inference is at the heart of many such approximate inference techniques. The key idea is to cast inference over an intractable joint distribution as an optimization problem over a family of tractable approximations to the true distribution [6, 34, 38]. Several such approximations exist, e.g., mean field, belief propagation etc. In this work, we focus on survey propagation. Inspired from

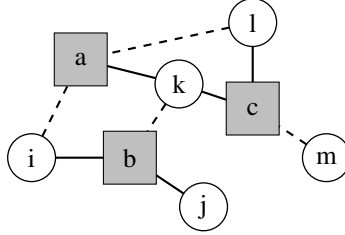

Figure 1: Factor graph for a 3-SAT instance with 5 variables (circles) and 3 clauses (squares). A solid (dashed) edge between a clause and a variable indicates that the clause contains the variable as a positive (negative) literal. This instance corresponds to $(\neg x_i \vee x_k \vee \neg x_l) \wedge (x_i \vee x_j \vee \neg x_k) \wedge (x_k \vee x_l \vee \neg x_m)$, with the clauses $a, b, c$ listed in order.

statistical physics, survey propagation is a message-passing algorithm that corresponds to belief propagation in a "lifted" version of the original CSP and underlines many state-of-the-art solvers for random CSPs [24, 22, 21].

Existing branching rules for survey propagation iteratively pick variables with the most confident marginals and fix their values (by adding unary constraints on these variables) in a process known as *decimation*. This heuristic works well in practice, but struggles with a high variance in the success of branching, as the unary constraints leave the *survey inspired decimation* algorithm unable to recover in the event that a contradictory assignment (i.e., one that cannot be completed to form a satisfying assignment) is made. Longer branching predicates, defined over multiple variables, have lower variance and are more effective both in theory and practice [14, 1, 2, 36, 19, 18].

In this work, we introduce improved branching heuristics for survey propagation by extending this idea to CSPs; namely, we show that branching on more complex predicates than single-variable constraints greatly improves survey propagation's ability to find solutions to CSPs. Appealingly, the more complex, multi-variable predicates which we refer to as *streamlining constraints*, can be easily implemented as additional factors (not necessarily unary anymore) in message-passing algorithms such as survey propagation. For this reason, branching on more complex predicates is a natural extension to survey propagation.

Using these new branching heuristics, we develop an algorithm and empirically benchmark it on families of random CSPs. Random CSPs exhibit sharp phase transitions between satisfiable and unsatisfiable instances and are an important model to analyze the average hardness of CSPs, both in theory and practice [25, 26]. In particular, we consider two such CSPs: $k$-SAT where constraints are restricted to disjunctions involving exactly $k$ (possibly negated) variables [3] and XORSAT which substitutes disjunctions in $k$-SAT for XOR constraints of fixed length. On both these problems, our proposed algorithm outperforms the competing survey inspired decimation algorithm that branches based on just single variables, increasing solver success rates.

## 2  Preliminaries

Every CSP can be encoded as a boolean SAT problem expressed in Conjunctive Normal Form (CNF), and we will use this representation for the remainder of this work. Let $V$ and $C$ denote index sets for $n$ Boolean *variables* and $m$ *clauses* respectively. A *literal* is a variable or its negation; a clause is a disjunction of literals. A CNF formula $F$ is a conjunction of clauses, and is written as $(l_{11} \vee \ldots \vee l_{1k_1}) \wedge \ldots \wedge (l_{m1} \vee \ldots \vee l_{mk_m})$. Each $(l_{j1} \vee \ldots \vee l_{jk_j})$ is a clause with $k_j$ literals. For notational convenience, the variables will be indexed with letters $i, j, k, \ldots$ and the clauses will be indexed with letters $a, b, c, \ldots$. Each variable $i$ is Boolean, taking values $x_i \in \{0, 1\}$. A formula is satisfiable is there exists an assignment to the variables such that all the clauses are satisfied, where a clause is satisfied if at least one literal evaluates to true.

Any SAT instance can be represented as an undirected graphical model where each clause corresponds to a factor, and is connected to the variables in its scope. Given an assignment to the variables in its scope, a factor evaluates to 1 if the corresponding clause evaluates to True, and 0 otherwise. The corresponding joint probability distribution is uniform over the set of satisfying assignments. An example factor graph illustrating the use of our notation is given in Figure 1.

$k$-SAT formulas are ones where all clauses $(l_{j1} \lor \ldots \lor l_{jk_j})$ have exactly $k$ literals, i.e., $k_j = k$ for $j = 1, \cdots, m$. Random $k$-SAT instances are generated by choosing each literal's variable and negation independently and uniformly at random in each of the $m$ clauses. It has been shown that these instances have a very distinctive behavior where the probability of an instance having a solution has a phase transition explained as a function of the constraint density, $\alpha = m/n$, for a problem with $m$ clauses and $n$ variables for large enough $k$. These instances exhibit a sharp crossover at a threshold density $\alpha_s(k)$: they are almost always satisfiable below this threshold, and they become unsatisfiable for larger constraint densities [12, 10]. Empirically, random instances with constraint density close to the satisfiability threshold are difficult to solve [23].

## 2.1 Survey propagation

The base algorithm used in many state-of-the-art solvers for constraint satisfaction problems such as random $k$-SAT is *survey inspired decimation* [7, 24, 16, 23]. The algorithm employs survey propagation, a message passing procedure that computes approximate single-variable marginal probabilities for use in a decimation procedure. Our approach uses the same message passing procedure, and we review it here for completeness.

Survey propagation is an iterative procedure for estimating variable marginals in a factor graph. In the context of a factor graph corresponding to a Boolean formula, these marginals represent approximately the probability of a variable taking on a particular assignment when sampling uniformly from the set of satisfying assignments of the formula. Survey propagation considers three kinds of assignments for a variable: $0$, $1$, or unconstrained (denoted by $*$). A high value for marginals corresponding to either of the first two assignments indicates that the variables assuming the particular assignment make it likely for the overall formula to be satisfiable, whereas a high value for the unconstrained marginal indicates that satisfiablility is likely regardless of the variable assignment.

In order to estimate these marginals from a factor graph, we follow a message passing protocol where we first compute *survey* messages for each edge in the graph. There are two kinds of survey messages: messages $\{\eta_{i \to a}\}_{i \in V, a \in C(i)}$ from variable nodes $i$ to clauses $a$, and messages $\{\eta_{a \to i}\}_{a \in C, i \in V(a)}$ from clauses to variables. These messages can be interpreted as *warnings of unsatisfiability*.

1. If we let $V(a)$ to be the set of variables appearing in clause $a$, then the message sent from a clause $a$ to variable $i$, $\eta_{a \to i}$, is intuitively the probability that all variables in $V(a) \backslash \{i\}$ are in the state that violates clause $a$. Hence, clause $a$ is issuing a warning to variable $i$.

2. The reverse message from variable $i$ to clause $a$ for some value $x_i$, $\eta_{i \to a}$, is interpreted as the probability of variable $i$ assuming the value $x_i$ that violates clause $a$.

As shown in Algorithm 1, the messages from factors (clauses) to variables $\eta_{a \to i}$ are initialized randomly [Line 2] and updated until a predefined convergence criteria [Lines 5-7]. Once the messages converge to $\eta_{a \to i}^*$, we can estimate the approximate marginals $\mu_i(0), \mu_i(1), \mu_i(*)$ for each variable $i$. In case survey propagation does not converge even after repeated runs, or a contradiction is found, the algorithm output is UNSAT. The message passing updates SP-Update [Line 6] and the marginalization procedure Marginalize [Line 9] are deferred to Appendix A for ease of presentation. We refer the reader to [24] and [7] for a detailed analysis of the algorithm and connections to statistical physics.

## 2.2 Decimation and Simplification

The magnetization of a variable $i$, defined as $M(i) := |\mu_i(0) - \mu_i(1)|$, is used as a heuristic bias to determine how constrained the variable is to take a particular value. The magnetization can be a maximum of one which occurs when either of the marginals is one and a minimum of zero when the estimated marginals are equal.[1] The decimation procedure involves setting the variable(s) with the highest magnetization(s) to their most likely values based on the relative magnitude of $\mu_i(0)$ vs. $\mu_i(1)$ [Lines 12-13].

The algorithm then branches on these variable assignments and simplifies the formula by unit propagation [Line 15]. In unit propagation, we recursively iterate over all the clauses that the decimated variable appears in. If the polarity of the variable in a literal matches its assignment, the clause is satisfied and hence, the corresponding clause node and all its incident variable edges are

**Algorithm 1** SurveyInspiredDecimation($V, C$)

---

1: Initialize $\mathcal{V} \leftarrow V$ and $\mathcal{C} \leftarrow C$
2: Initialize messages $\{\eta_{a \rightarrow i}\}_{a \in C, i \in V(a)}$ at random
3: **while** $(\sum_i |\mu_i(0) - \mu_i(1)| > \epsilon)$ **do**
4:     ▷ Message passing inference
5:    **repeat**
6:       $\{\eta_{a \rightarrow i}\} \leftarrow$ SP-Update($\mathcal{V}, \mathcal{C}, \{\eta_{a \rightarrow i}\}$)
7:    **until** Convergence to $\{\eta_{a \rightarrow i}^*\}$
8:    **for** $i = 1, \ldots, |\mathcal{V}|$ **do**
9:       $\mu_i(0), \mu_i(1), \mu_i(*) \leftarrow$ Marginalize($\mathcal{V}, \mathcal{C}, \{\eta_{a \rightarrow i}\}$)
10:   **end for**
11:   ▷ Branching (Decimation)
12:   Choose $i^* \leftarrow \arg\max_{i \in \mathcal{V}} |\mu_i(0) - \mu_i(1)|$
13:   Set $y^* \leftarrow \arg\max_{y \in \{0,1\}} \mu_{i^*}(y)$
14:   ▷ Simplification
15:   Update $\mathcal{V}, \mathcal{C} \leftarrow$ UnitPropagate($\mathcal{V}, \mathcal{C} \cup \{x_{i^*} = y^*\}$)
16: **end while**
17: **return** LocalSearch($\mathcal{V}, \mathcal{C}$)

---

removed from the factor graph. If the polarity in the literal does not match the assignment, only the edge originating from this particular variable node incident to the clause node is removed from the graph. For example, setting variable $k$ to 0 in Figure 1 leads to removal of edges incident to $k$ from $a$ and $c$, as well as all outgoing edges from $b$ (because $b$ is satisfied).

## 2.3 Survey Inspired Decimation

The full iterative process of survey propagation (on the simplified graph from the previous iteration) followed by decimation is continued until a satisfying assignment is found, or a stopping condition is reached beyond which the instance is assumed to be sufficiently easy for local search using a standard algorithm such as WalkSAT [31]. Note that when the factor graph is a tree and survey propagation converges to the exact warning message probabilities, Algorithm 1 is guaranteed to select good variables to branch on and to find a solution (assuming one exists).

However, the factor graphs for CSPs are far from tree-like in practice and thus, the main factor affecting the success of survey inspired decimation is the quality of the estimated marginals. If these estimates are inaccurate, it is possible that the decimation procedure chooses to fix variables in contradictory configurations. To address this issue, we propose to use streamlining constraints.

# 3 Streamlining survey propagation

Combinatorial optimization algorithms critically depend on good heuristics for deciding where to branch during search [5]. Survey propagation provides a strong source of information for the decimation heuristic. As discussed above, the approximate nature of message-passing implies that the "signal" might be misleading. We now describe a more effective way to use the information from survey propagation.

Whenever we have a combinatorial optimization problem over $X = \{0, 1\}^n$ and wish to find a solution $s \in S \subseteq X$, we may augment the original feasibility problem with constraints that partition the statespace $X$ into disjoint statespaces and recursively search the resulting subproblems. Such partitioning constraints can significantly simplify search by exploiting the structure of the solution set $S$ and are known as *streamlining constraints* [17]. Good streamlining constraints will provide a balance between yielding significant shrinkage of the search space and safely avoiding reductions in the solution density of the resulting subproblems. Partitioning the space based on the value of a single variable (like in decimation) performs well on the former at the cost of the latter. We therefore introduce a different constraining strategy that strives to achieve a more balanced trade-off.

## 3.1 Streamlining constraints for constraint satisfaction problems

The success of survey inspired decimation relies on the fact that marginals carry some signal about the likely assignments of variables. However, the factor graph becomes more dense as the constraint density approaches the phase transition threshold, making it harder for survey propagation to converge in practice. This suggests that the marginals might provide a weaker signal to the decimation procedure in early iterations. Instead of selecting a variable to freeze in some configuration as in decimation, *e.g.*, $x_i = 1$, we propose a strictly more general streamlining approach where we use *disjunction constraints* between subsets of highly magnetized variables, *e.g.*, $(x_i \vee x_j) = 1$.

The streamlined constraints can cut out smaller regions of the search space while still making use of the magnetization signal. For instance, introducing a disjunction constraint between any pair of variables reduces the state-space by a factor of $4/3$ (since three out of four possible variable assignments satisfy the clause), in contrast to the decimation procedure in Algorithm 1 which reduces the state space by a factor of 2. Intuitively, when branching with a length-2 clause such as $(x_i \vee x_j)$ we make an (irreversible) mistake only if we guess the value of *both* variables wrong. Decimation can also be seen as a special case of streamlining for the same choice of literal. To see why, we note that in the above example the acceptable variable assignments for decimation $(x_i, x_j) = \{(1, 0), (1, 1)\}$ are a subset of the valid assignments for streamlining $(x_i, x_j) = \{(1, 0), (1, 1), (0, 1)\}$.

The success of the streamlining constraints is strongly governed by the literals selected for participating in these added disjunctions. Disjunctions could in principle involve any number of literals, and longer disjunctions result in more conservative branching rules. But there are diminishing returns with increasing length, and so we restrict ourselves to disjunctions of length at most two in this paper. Longer clauses can in principle be handled by the inference procedure used by message-passing algorithms, and we leave an exploration of this extension to future work.

## 3.2 Survey Inspired Streamlining

The pseudocode for survey inspired streamlining is given in Algorithm 2. The algorithm replaces the decimation step of survey inspired decimation with a streamlining procedure that adds disjunction constraints to the original formula [Line 16], thereby making the problem increasingly constrained until the search space can be efficiently explored by local search.

For designing disjunctions, we consider candidate variables with the highest magnetizations, similar to decimation. If a variable $i$ is selected, the polarity of the literal containing the variable is positive if $\mu_i(1) > \mu_i(0)$ and negative otherwise [Lines 12-15].

Disjunctions use the signal from the survey propagation messages without overcommitting to a particular variable assignment too early (as in decimation). Specifically, without loss of generality, if we are given marginals $\mu_i(1) > \mu_i(0)$ and $\mu_j(1) > \mu_j(0)$ for variables $i$ and $j$, the new update adds the streamlining constraint $x_i \vee x_j$ to the problem instead of overcommitting by constraining $i$ or $j$ to its most likely state. This approach leverages the signal from survey propagation, namely that it is unlikely for $\neg x_i \wedge \neg x_j$ to be true, while also allowing for the possibility that one of the two marginals may have been estimated incorrectly. As long as streamlined constraints and decimation use the same bias signal (such as magnetization) for ranking candidate variables, adding streamlined constraints through the above procedure is guaranteed to not degrade performance compared with the decimation strategy in the following sense.

**Proposition 1.** *Let $F$ be a formula under consideration for satisfiability, $F_d$ be the formula obtained after one round of survey inspired decimation, and $F_s$ be the formula obtained after one round of survey inspired streamlining. If $F_d$ is satisfiable, then so is $F_s$.*

*Proof.* Because unit-propagation is sound, the formula obtained after one round of survey inspired decimation is satisfiable if and only if $(F \wedge \ell_{i^*})$ is satisfiable, where the literal $\ell_{i^*}$ denotes either $x_{i^*}$ or $\neg x_{i^*}$. By construction, the formula obtained after one round of streamlining is $F \wedge (\ell_{i^*} \vee \ell_{j^*})$. It is clear that if $(F \wedge \ell_{i^*})$ is satisfiable, so is $F \wedge (\ell_{i^*} \vee \ell_{j^*})$. Clearly, the converse need not be true. $\square$

## 3.3 Algorithmic design choices

A practical implementation of survey inspired streamlining requires setting some design hyperparameters. These hyperparameters have natural interpretations as discussed below.

---

**Algorithm 2** SurveyInspiredStreamlining($V, C, T$)

---
1: Initialize $\mathcal{V} \leftarrow V$ and $\mathcal{C} \leftarrow C$
2: Initialize messages $\{\eta_{a \rightarrow i}\}_{a \in C, i \in V(a)}$ at random
3: **while** $\sum_i |\mu_i(0) - \mu_i(1)| \geq \epsilon$ **do**
4:     **repeat**
5:         $\{\eta_{a \rightarrow i}\} \leftarrow$ SP-Update($\mathcal{V}, \mathcal{C}, \{\eta_{a \rightarrow i}\}$)
6:     **until** Convergence to $\{\eta^*_{a \rightarrow i}\}$
7:     **for** $i = 1, \ldots, |\mathcal{V}|$ **do**
8:         $\mu_i(0), \mu_i(1), \mu_i(*) \leftarrow$ Marginalize($\mathcal{V}, \mathcal{C}, \{\eta_{a \rightarrow i}\}$)
9:     **end for**
10:    **if** $t < T$ **then**
11:        ▷ Add Streamlining Constraints
12:        Choose $i^* \leftarrow \arg\max_{i \in \mathcal{V}} |\mu_i(0) - \mu_i(1)|$
13:        Choose $j^* \leftarrow \arg\max_{i \in \mathcal{V}, i \neq i^*} |\mu_i(0) - \mu_i(1)|$
14:        Set $y^* \leftarrow \arg\max_{y \in \{0,1\}} \mu_{i^*}(y)$
15:        Set $w^* \leftarrow \arg\max_{y \in \{0,1\}} \mu_{j^*}(y)$
16:        $\mathcal{C} \leftarrow \mathcal{C} \cup \{x_{i^*} = y^* \vee x_{j^*} = w^*\}$
17:    **else**
18:        Choose $i^* \leftarrow \arg\max_{i \in \mathcal{V}} |\mu_i(0) - \mu_i(1)|$
19:        Set $y^* \leftarrow \arg\max_{y \in \{0,1\}} \mu_{i^*}(y)$
20:        $\mathcal{V}, \mathcal{C} \leftarrow$ UnitPropagate($\mathcal{V}, \mathcal{C} \cup \{x_{i^*} = y^*\}$)
21:    **end if**
22: **end while**
23: **return** LocalSearch($\mathcal{V}, \mathcal{C}$)

---

**Disjunction pairing.** Survey inspired decimation scales to large instances by taking the top $R$ variables as decimation candidates at every iteration instead of a single candidate (Line 13 in Algorithm 1). The parameter $R$ is usually set as a certain fraction of the total number of variables $n$ in the formula, e.g., $1\%$. For the streamlining constraints, we take the top $2 \cdot R$ variables, and pair the variables with the highest and lowest magnetizations as a disjunction constraint. We remove these variables from the candidate list, repeating until we have added $R$ disjunctions to the original set of constraints. For instance, if $v_1, \cdots, v_{2R}$ are our top decimation candidates (with signs) in a particular round, we add the constraints $(v_1 \vee v_{2R}) \wedge (v_2 \vee v_{2R-1}) \wedge \cdots \wedge (v_R \vee v_{R+1})$. Our procedure for scaling to top $R$ decimation candidates ensures that Proposition 1 holds, because survey inspired decimation would have added $(v_1) \wedge (v_2) \wedge \cdots \wedge (v_R)$ instead.

Other pairing mechanisms are possible, such as for example $(v_1 \vee v_{R+1}) \wedge (v_2 \vee v_{R+2}) \wedge \cdots \wedge (v_R \vee v_{R+R})$. Our choice is motivated by the observation that $v_{2R}$ is the variable we are least confident about - we therefore choose to pair it with the one we are most confident about ($v_1$). We have found our pairing scheme to perform slightly better in practice.

**Constraint threshold.** We maintain a streamlining constraint counter for every variable which is incremented each time the variable participates in a streamlining constraint. When the counter reaches the constraint threshold, we no longer consider it as a candidate in any of the subsequent rounds. This is done to ensure that no single variable dominates the constrained search space.

**Iteration threshold.** The iteration threshold $T$ determines how many rounds of streamlining constraints are performed. While streamlining constraints smoothly guide search to a solution cluster, the trade-off being made is in the complexity of the graph. With every round of addition of streamlining constraints, the number of edges in the graph increases which leads to a higher chance of survey propagation failing to converge. To sidestep the failure mode, we perform $T$ rounds of streamlining before switching to decimation.

## 4  Empirical evaluation

We streamlining constraints for random $k$-SAT instances for $k = \{3, 4, 5, 6\}$ with $n = \{5 \times 10^4, 4 \times 10^4, 3 \times 10^4, 10^4\}$ variables respectively and constraint densities close to the theoretical predictions of the phase transitions for satisfiability.

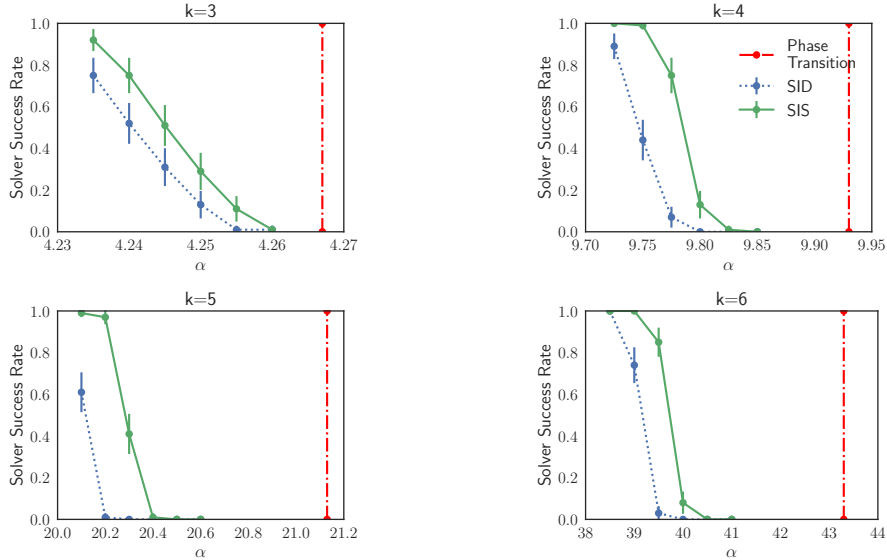

Figure 2: Random $k$-SAT solver rates (with 95% confidence intervals) for $k \in \{3, 4, 5, 6\}$, for varying constraint densities $\alpha$. The red line denotes the theoretical prediction for the phase transition of satisfiability. Survey inspired streamlining (SIS) drastically outperforms survey inspired decimation (SID) for all values of $k$.

## 4.1 Solver success rates

In the first set of experiments, we compare survey inspired streamlining (SIS) with survey inspired decimation (SID). In line with [7], we fix $R = 0.01n$ and each success rate is the fraction of 100 instances solved for every combination of $\alpha$ and $k$ considered. The constraint threshold is fixed to 2. The iteration threshold $T$ is a hyperparameter set as follows. We generate a set of 20 random $k$-SAT instances for every $\alpha$ and $k$. For these 20 "training" instances, we compute the empirical solver success rates varying T over $\{10, 20, \ldots, 100\}$. The best performing value of $T$ on these train instances is chosen for testing on 100 fresh instances. All results are reported on the test instances.

**Results.** As shown in Figure 2, the streamlining constraints have a major impact on the solver success rates. Besides the solver success rates, we compare the algorithmic thresholds which we define to be the largest constraint density for which the algorithm achieves a success rate greater than 0.05. The algorithmic thresholds are pushed from 4.25 to 4.255 for $k = 3$, 9.775 to 9.8 for $k = 4$, 20.1 to 20.3 for $k = 5$, and 39 to 39.5 for $k = 6$, shrinking the gap between the algorithmic thresholds and theoretical limits of satisfiability by an average of 16.3%. This is significant as there is virtually no performance overhead in adding streamlining constraints.

**Distribution of failure modes.** Given a satisfiable instance, solvers based on survey propagation could fail for two reasons. First, the solver could fail to converge during message passing. Second, the local search procedure invoked after simplification of the original formula could timeout which is likely to be caused due to a pathological simplification that prunes away most (or even all) of the solutions. In our experiments, we find that the percentage of failures due to local search timeouts in SID and SIS are 36% and 24% respectively (remaining due to non-convergence of message passing).

These observations can be explained by observing the effect of decimation and streamlining on the corresponding factor graph representation of the random $k$-SAT instances. Decimation simplifies the factor graph as it leads to the deletion of variable and factor nodes, as well as the edges induced by the deleted nodes. This typically reduces the likelihood of non-convergence of survey propagation since the graph becomes less "loopy", but could lead to overconfident (incorrect) branching decisions especially in the early iterations of survey propagation. On the other hand, streamlining takes smaller steps in reducing the search space (as opposed to decimation) and hence are less likely to make inconsistent variable assignments. However, a potential pitfall is that these constraints add factor nodes that make the graph more dense, which could affect the convergence of survey propagation.

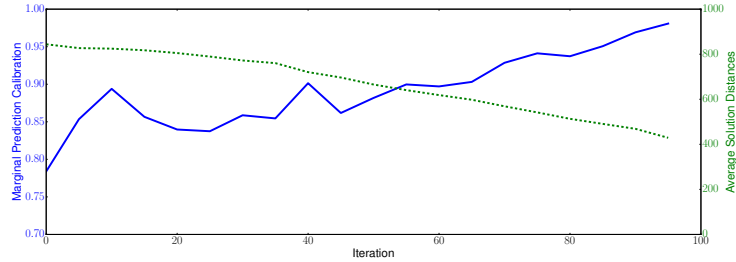

Figure 3: Marginal prediction calibration (blue) and sampled solution distances (green) during solver run on 3-SAT with $5000$ variables, $\alpha = 4.15$, $T = 90$.

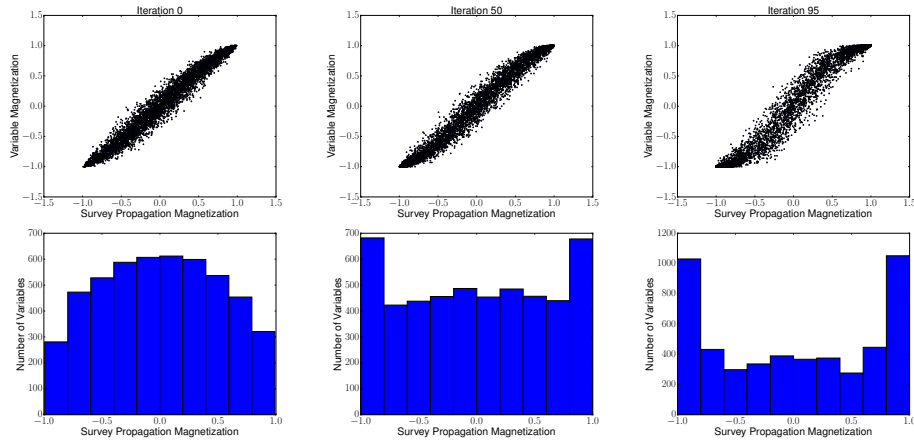

Figure 4: **Top:** Correlation between magnetization and estimated marginal probabilities for the same problem instance as we add streamlining constraints. **Bottom:** Histogram of variables magnetizations. As streamlining constraints are added, the average confidence of assignments increases.

## 4.2 Solution cluster analysis

Figures 3 and 4 reveal the salient features of survey inspired streamlining as it runs on an instance of 3-SAT with a constraint density of $\alpha = 4.15$, which is below the best achievable density but is known to be above the clustering threshold $\alpha_d(3) \approx 3.86$. The iteration threshold, $T$ was fixed to 90. At each iteration of the algorithm we use SampleSAT [35] to sample 100 solutions of the streamlined formula. Using these samples we estimate the marginal probabilities of all variables *i.e.*, the fraction of solutions where a given variable is set to true. We use these marginal probabilities to estimate the *marginal prediction calibration i.e.*, the frequency that a variable which survey propagation predicts has magnetization at least 0.9 has an estimated marginal at least as high as the prediction.

The increase in marginal prediction calibrations during the course of the algorithm (Figure 3, blue curve) suggests that the streamlining constraints are selecting branches that preserve most of the solutions. This might be explained by the decrease in the average Hamming distance between pairs of sampled solutions over the course of the run (green curve). This decrease indicates that the streamlining constraints are guiding survey propagation to a subset of the full set of solution clusters.

Over time, the algorithm is also finding more extreme magnetizations, as shown in the bottom three histograms of Figure 4 at iterations 0, 50, and 95. Because magnetization is used as a proxy for how reliably one can branch on a given variable, this indicates that the algorithm is getting more and more confident on which variables it is "safe" to branch on. The top plots of Figure 4 show the empirical marginal of each variable versus the survey propagation magnetization. These demonstrate that overall the survey propagation estimates are becoming more and more risk-averse: by picking variables with high magnetization to branch on, it will only select variables with (estimated) marginals close to one.

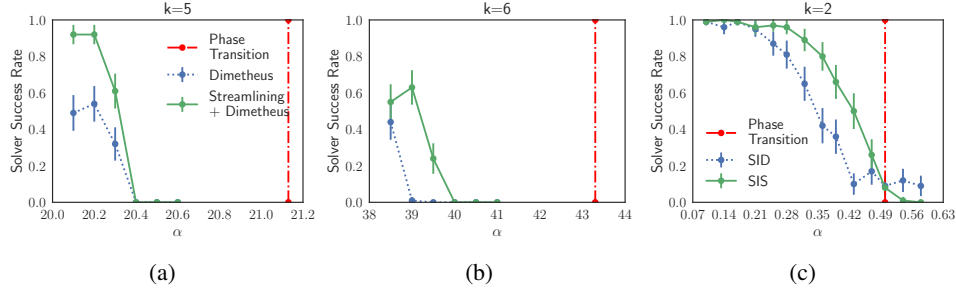

Figure 5: (a, b) Random $k$-SAT solver rates (with $95\%$ confidence intervals) for $k \in \{5, 6\}$ testing integration with Dimetheus. (c) XORSAT solver rates (with $95\%$ confidence intervals).

### 4.3 Integration with downstream solvers

The survey inspired streamlining algorithm provides an easy "black-box" integration mechanism with other solvers. By adding streamlining constraints in the first few iterations as a preprocessing routine, the algorithm carefully prunes the search space and modifies the original formula that can be subsequently fed to any external downstream solver. We tested this procedure with Dimetheus [16] – a competitive ensemble solver that won two recent iterations of the SAT competitions in the random $k$-SAT category. We fixed the hyperparameters to the ones used previously. We did not find any statistically significant change in performance for $k = 3, 4$; however, we observe significant improvements in solver rates for higher $k$ (Figure 5a, 5b).

### 4.4 Extension to other constraint satisfaction problems

The survey inspired streamlining algorithm can be applied to any CSP in principle. Another class of CSPs commonly studied is XORSAT. An XORSAT formula is expressed as a conjunction of XOR constraints of a fixed length. Here, we consider constraints of length 2. An XOR operation $\oplus$ between any two variables can be converted to a conjunction of disjunctions by noting that $x_i \oplus x_j = (\neg x_i \vee \neg x_j) \wedge (x_i \vee x_j)$, and hence, any XORSAT formula can be expressed in CNF form. Figure 5c shows the improvements in performance due to streamlining. While we note that the phase transition is not as sharp as the ones observed for random $k$-SAT (in both theory and practice [11, 28]), including streamlining constraints can improve the solver performance.

## 5 Conclusion

Variational inference algorithms based on survey propagation achieve impressive performance for constraint satisfaction problems when employing the decimation heuristic. We explored cases where decimation failed, motivating a new branching procedure based on streamlining constraints over disjunctions of literals. Using these constraints, we developed survey inspired streamlining, an improved algorithm for solving CSPs via variational approximations. Empirically, we demonstrated improvements over the decimation heuristic on random CSPs that exhibit sharp phase transitions for a wide range of constraint densities. Our solver is available publicly at https://github.com/ermongroup/streamline-vi-csp.

An interesting direction for future work is to integrate streamlining constraints with backtracking. Backtracking expands the search space, and hence it introduces a computational cost but typically improves statistical performance. Similar to the backtracking procedure proposed for decimation [23], we can backtrack (delete) streamlining constraints that are unlikely to render the original formula satisfiable during later iterations of survey propagation. Secondly, it would be interesting to perform survey propagation on clusters of variables and to use the joint marginals of the clustered variables to decide which streamlining constraints to add. The current approach makes the simplifying assumption that the variable magnetizations are independent of each other. Performing survey propagation on clusters of variables could greatly improve the variable selection while incurring only a moderate computational cost. Finally, it would be interesting to extend the proposed algorithm for constraint satisfaction in several real-world applications involving combinatorial optimization such as planning, scheduling, and probabilistic inference [20, 8, 9, 39, 37].

## Acknowledgments

This research was supported by NSF (#1651565, #1522054, #1733686) and FLI. AG is supported by a Microsoft Research PhD Fellowship and a Stanford Data Science Scholarship. We are grateful to Neal Jean for helpful comments on early drafts.

## Footnotes

[1]Other heuristic biases are also possible. For instance, [23] use the bias $1 - \min(\mu_i(1), \mu_i(0))$.

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
