[Supplementary Material · supplementary.pdf]

# Appendices

## A  Survey propagation subroutines

Here, we provide details regarding the subroutines regarding the message passing updates SP-Update [Line 6] and the marginalization procedure Marginalize [Line 9] used in Algorithms 1 and 2.

---

**Algorithm 3** SP-Update$(V, C, \{\eta_{a \to i}\})$

---

1: **for all** $a \in C, i \in V(a)$ **do**
2:     **for all** $j \in V(a) \backslash i$ **do**

3:
$$\eta_{j \to a}^{u} \leftarrow \left[1 - \prod_{b \in C_a^u(j)} (1 - \eta_{a \to j})\right] \prod_{b \in C_a^s(j)} (1 - \eta_{a \to j})$$

4:
$$\eta_{j \to a}^{s} \leftarrow \left[1 - \prod_{b \in C_a^s(j)} (1 - \eta_{a \to j})\right] \prod_{b \in C_a^u(j)} (1 - \eta_{a \to j})$$

5:
$$\eta_j^0 \leftarrow \prod_{b \in C(j) \backslash a} (1 - \eta_{a \to j})$$

6:     **end for**
7:     ▷ Compute new message
8:     $\eta'_{a \to i} \leftarrow \prod_{j \in V(a) \backslash i} \frac{\eta_{j \to a}^u}{\eta_{j \to a}^u + \eta_{j \to a}^s + \eta_j^0}$
9: **end for**
10: **return** $\{\eta'_{a \to i}\}$

---

### A.1  SP-Update

If we let $C_a^s(i)$ to be the set of clauses where $i$ appears with the same sign as in clause $a$ and $C_a^u(i)$ to be the remaining clauses, then the subroutine in Algorithm 3 provides the message passing equations required to update $\eta_{a \to i}$.

### A.2  Marginalize

We can estimate the approximate marginals $\mu_i(0), \mu_i(1), \mu_i(*)$ for each variable $i$ by normalizing the following quantities so that they sum to one:

$$\mu_i(1) \propto \left[1 - \prod_{a \in C_-(i)} (1 - \eta_{a \to i}^*)\right] \prod_{a \in C_+(i)} (1 - \eta_{a \to i}^*) \tag{1}$$

$$\mu_i(0) \propto \left[1 - \prod_{a \in C_+(i)} (1 - \eta_{a \to i}^*)\right] \prod_{a \in C_-(i)} (1 - \eta_{a \to i}^*) \tag{2}$$

$$\mu_i(*) \propto \prod_{a \in C(i)} (1 - \eta_{a \to i}^*) \tag{3}$$

where $C(i)$ denotes the set of clauses that $i$ appears in and $C_-(i)$ and $C_+(i)$ are the clause subsets where $i$ appears negated and unnegated respectively.