[Reviews · NeurIPS 2018]

Reviewer 1



The work proposes the use of streamlining in the context of survey inspired decimation algorithms---a main approach alongside stochastic local search---for effciiently finding solutions to large satisfiable random instances of the Boolean satisfiability (SAT) problem. The paper is well-written and easy to follow (although some hasty mistakes remain, see below). The proposed approach is shown to improve the state of the art (to some extent) in algorithms for solving random k-SAT instances, especially by showing that streamlining constraints allow for solving instances that a closer to the sat-unsat phase transition point than previously for different values of k. In terms of motivations, while I do find it of interest to develop algorithmic approach which allow for more efficiently finding solutions to the hardest random k-SAT instances, it would be beneficial if the authors would expand the introduction with more motivations for the work. In terms of contributions, the proposal consists essentially of combining previous proposed ideas to obtain further advances (which is of course ok, but slightly lowers the novelty aspects). The proposed changes are not very deep, but the empirical results are quite good and the experiments seem well-performed. I find it especially interesting that the state-of-the-art Dimetheus solver benefits from the proposed idea. More detailed comments: Lines 48-49: The claim is not totally correct; DPLL and CDCL algorithm could in principle easily be made to branch on more complex Boolean combinations, the challenge is more in coming up with effective heuristics for doing this. Line 81: "most" solvers is claiming too much. More traditional stochastic local search is also a n active area of development, and it is not entirely clear which approach currently prevails and when. In fact, the newest results from the SAT Competition will be available in a week or so, and the authors can check the situation and perhaps comment on this in the paper. http://sat2018.forsyte.tuwien.ac.at Line 107-108: reference are broken! Section 3.1: The authors should comment on the claim that increasing length of streamlining constraints is a "straightforward extension". The number of such constraints is exponential in the length, I to me it is not at all evident how to go about choosing from such an exponential set when increasing the length? I wonder if there are more practical reasons for restricting to length 2 ... In the beginning of Section 4.2 it would be good to explicate whether the alpha reached is the best one that has ever been reached, or has this been done before? Line 239: $T$ Many of the plots are semi-unreadable on a black-and-white printout. The resolution of the images should be improved and the plots made larger (or otherwise more readable). As a final comment, I would encourage the authors to submit the solver to the random SAT track of the next edition of the SAT Competitions. AFTER AUTHOR RESPONSE: Thank you for the constructive author response. I maintain my positive view on the work. It will be very interesting to see the results for Sparrow2Riss.

Reviewer 2



In this work, a new branching strategy for survey propagation is proposed. Specifically, the paper introduce 'streamlining constraint' in place of using decimation, which adds disjunction contraints between subsets of highly magnetized variables. The algorithm is empirically reported to improve existing k-SAT solvers. In overall, the paper was easy to read and idea seemed novel to me. I liked how the extension from decimation to streamline was very simple, but possess the potential to provoke further related works. My only concern on the scheme itself is that adding streamlining constraints will increase the size of problem (as mentioned briefly in paper), and reporting how much the algorithm is slower than just using decimation would make the paper more useful for practical applications. Minor comments: - In algorithmic design choices, it seems rather natural to use memory threshold in place of iteration threshold, since it is a more closely related quantity to our computational resources. For example, one could think of a scheme which combines streamlining and decimation by adding streamlining constraint whenever memory is possible, and using decimation otherwise. - In context of MAP inference in undirected graphical models (GMs), there is a similar scheme that is called 'uprooting and rerooting', which modifies the GM while keeping the MAP assignment invariant. It would be very interesting to see whether if similar scheme is effective in max-product belief propagation for GMs. --- I have read the author's reponse and considered it in my review.

Reviewer 3



The article proposes a method for adding streamlining constraints in the survey propagation procedure, which improves its performance on random k-SAT problems. In line with classical survey propagation, they use a message passing procedure to compute approximate marginal. However instead of using the marginal to set single variables in the decimation process as previously done in survey propagation, the authors propose to add constraints based upon multiple variables. Iterating this procedure, more highly constrained problems are produced, which can then be solved by a downstream solver. The authors show that this idea improved upon a state-of-the-art survey propagation solver. The article is well written, presents a novel and effective idea that I expect to see used in future SAT-competitions. While perhaps needed for the target audience of NIPS, I personally find the background section where previous algorithms are presented a bit too long. A substantial part of the paper is dedicated to the background that the reader could have easily been directed to by citations. I have reviewed this paper at UAI 2017. I voted for acceptance at that time, and I am still leaning towards accepting this paper. One reviewer at UAI 2017 raised highly valid points, which is that incorrect and highly imprecise claims are made in several parts of the paper. I am very glad to see that the authors at least made some effort in revising these overclaiming statements. For example, now the authors only claim that their solvers outperform decimation-only solvers for random k-SAT instances in the abstract, instead of saying outperforming all solvers. There are still a few places that I am going to bring authors’ attention. In particular, the authors should not make an implication that their method is analogous to streamlining random constraints in model counting, as in [2, 3, 14]. Agreed with reviewer 2 from UAI 2017, I think the contexts are entirely different, and the counting approaches in [2, 3, 14] simply cannot work with very short constraints, such as length-2 constraints used in this paper. I really think the authors should revise their statements in the third paragraph of the introduction. Once this paper is converted to NIPS style, Figure 2 becomes way too small. Please enlarge the fonts in Figure 2. In addition, below are my detailed comments from UAI 2017. On the technical side, there are a few directions that are not explored as much as would be ideal. First, the streamlining clauses are added based on the two most polarized variables. As an alternative, we can define a slightly different message passing algorithm which computes the marginal of the disjunction of all pairwise variables. Then we add streamlining constraints based on the pairwise marginal, instead of selecting the two most polarized variables assuming the independence. Will this be a better method? The algorithmic design choices of section 3.3 could have been motivated more clearly. In particular, it would be interesting to plot how different choices affect the performance. Secondly, the constraints are just added to a single solver which is treated like a black box. To make the case of generality for the method, it would be preferable to use more solvers or have a metric of success that doesn’t depend on the selected black-box solver. There are also a few presentation details that should be addressed. In section 3.3 the reader is directed to a theorem 1 which does not exist, I assume that it is supposed to be proposition 1. It is assumed that the three rows of figure 3 related to the same run - that could however be made clearer in the caption. In the introduction, only very recent work related to the phase transition of k-SAT is referred to. It would be good to cite several original publications.